# Effect of Zeolite on Shrinkage and Crack Resistance of High-Performance Cement-Based Concrete

**DOI:** 10.3390/ma13173773

**Published:** 2020-08-26

**Authors:** Nguyen Cong Thang, Nguyen Van Tuan, Keun-Hyeok Yang, Quoc Tri Phung

**Affiliations:** 1Faculty of Building Material, National University of Civil Engineering, Hanoi 100000, Vietnam; thangnc@nuce.edu.vn; 2Department of Architectural Engineering, Kyonggi University, Suwon 16227, Korea; yangkh@kgu.ac.kr; 3Institute for Environment, Health, and Safety, Belgian Nuclear Research Centre (SCK CEN), 2400 Mol, Belgium; quoc.tri.phung@sckcen.be

**Keywords:** high-performance concrete, autogenous shrinkage, cracking, zeolite, *fib* 2010 model

## Abstract

This study examined the effectiveness of zeolite addition to reduce the autogenous shrinkage of high-performance cement-based concrete (HPC). The zeolites were replaced up to 15% of the cement content by weight and their mean particle size varied from 5.6 to 16.7 µm. To evaluate the crack resistance of HPC containing zeolites, the ring tests and internal relative humidity measurements were performed at different ages. The compressive strengths were determined at 3, 7, 28 and 90 days of curing. Test results confirmed that the addition of zeolite was promising and favourable in enhancing the compressive strength, crack resistance and reducing the autogenous shrinkage of HPC due to synergistic pozzolanic and internal curing effects. The autogenous shrinkage tended to decrease with the increase in zeolite content and its particle size. In addition, the extent of the autogenous shrinkage development at the early ages decreased with higher zeolite content replaced. Furthermore, to predict the autogenous shrinkage of HPC containing zeolite, an improved model has been proposed, in which the conventional ultimate autogenous shrinkage strain and time function were modified by introducing new parameters accounting for the zeolite content and its particle size. It appeared that the proposed model was able to capture the autogenous shrinkage behaviour of HPC with or without zeolite, while the *fib* 2010 model underestimated the autogenous shrinkage of HPC containing less than 10% zeolite replacement.

## 1. Introduction

High-performance cement-based concretes (HPC), such as high-performance concrete and ultra-high-performance concrete, have been considered as promising materials in the last decade because of their advanced characteristics in the compressive and tensile strength, stiffness and long-term durability compared to the normal or high-strength concretes. Such concretes typically have low water-to-binder (w/b) ratio and high binder content, including cement and mineral admixtures, which results in a high compressive strength, high elastic modulus and excellent durability. However, a large amount of binder in the matrix could, consequently, cause a disadvantage not only in economic efficiency but also in engineering properties under working conditions. Commonly, HPC exhibits a larger shrinkage compared to normal concrete and autogenous shrinkage significantly contributes to the total shrinkage in HPC compared to drying shrinkage. In the first few days after mixing, HPC already shows a considerably high autogenous shrinkage, which increases the potential of cracking at early ages. The risk of cracking induced by the restrained autogenous shrinkage would limit the application of HPC in construction. Furthermore, HPC offers an extremely dense matrix, however, with a few cracks appearing in the HPC, the transport properties of concrete (e.g., permeability or diffusion [1]) would significantly increase, which reduces the durability of HPC. Basically, the large autogenous shrinkage of HPC is originated from the extremely low water/binder ratio and remarkably high silica fume (SF) content normally used in HPC. Therefore, the internal relative humidity (RH) in the cementitious matrix is significantly dropped during hardening of concrete. Furthermore, self-desiccation also occurs under some sealed curing conditions (i.e., without an external source of water) [2], which induces further RH drop.

Shrinkage deformation is a very complicated phenomenon which generates tensile stress and can cause structural cracking when it is restrained with reinforcement and formwork, adversely affecting the properties of concrete such as strength, permeability, durability, etc. [3,4]. It should be noted that for HPC with a dense structure, mitigation of autogenous shrinkage using external curing is not effective because of its extremely low permeability, which limits the water ingress into the concrete components. For this reason, one of the effective solutions to counteract self-desiccation and autogenous shrinkage of HPC (with low w/b ratio) and thereby potential of early-age cracking is the use of internal water curing approach. The most popular method applied is the use of water-saturated porous aggregates including zeolites [5,6,7], rice husk ash [8,9,10], porous pumice [11] and super-absorbent polymers (SAPs) [12,13], of which the internal curing capacity enables to compensate the shrinkage in concrete [14,15,16]. Besides, the cement amount could be reduced by using mineral admixtures (to partially replace cement), which also helps to increase the sustainability of concrete [17]. The question arising at this point is whether a single material can play “a duplex role” which acts as not only a mineral admixture but also as an internal water curing agent for concrete. If such materials are available, numerous advantages in production of HPC could be achieved. This concern is the focus of this research by using zeolites, which have potential to play a duplex role, to reduce the autogenous shrinkage of HPC. The internal curing effect is understood as a process that promotes the hydration of cement and prevents the loss of water in concrete. Water can be absorbed or released from the saturated zeolite and hence, retain the internal relative humidity of concrete, resulting in the shrinkage compensation and also the hydration enhancement [7]. It should be noted that curing not only aims to increase the strength of concrete but also to improve the dense structure, reduce permeability and enhance durability for concrete. However, when using zeolite with a large quantity, the absorbed water for zeolite increases rapidly, leading to a decrease in the amount of water required for early age hydration (first few hours). Therefore, the workability of the concrete mixture decreases, and this often leads to a high required dosage of superplasticizer to ensure the workability of the concrete mixture [18]. Though the addition of zeolite results in a mitigation of autogenous shrinkage of HPC, a slight reduction of compressive strength of HPC at all ages might occur, especially with a high zeolite replacement [19,20].

In the context of limited experimental data available for autogenous shrinkage of HPC with low water/binder ratio [7,19,20], there is a need to further explore the potential of shrinkage of HPC, especially with the addition of shrinkage reducing agents. This study intensively examined the effects of zeolite addition on the shrinkage of HPC. To ascertain the crack resistance of HPC with zeolite, compressive strength development and restrained shrinkage behaviour were evaluated at different ages on HPC with various zeolite replacement levels and particle sizes. Furthermore, an empirical model to access the evolution of shrinkage over time was proposed based on the regression analysis using experimental data to evaluate the autogenous shrinkage behaviour of HPC containing zeolite, which was then compared with the existing *fib* 2010 model [21].

## 2. Materials and Methods

### 2.1. Materials

The materials used in this study included quartz sand (S) with a mean particle size of about 300 μm, and Portland cement PC40 (C) (complying with Vietnamese Standard TCVN 2682: 2009 [22]). Zeolites (Ze) were employed as a partial replacement of cement to reduce the autogenous shrinkage of HPC. The effects of particles size were investigated by using zeolites with 3 mean particle sizes of 5.6, 10.3 and 16.7 µm (abbreviated as Ze-5.6, Ze-10.3, and Ze-16.7, respectively), measured by laser diffraction. The density of cement and zeolite were determined according to Vietnamese Standard TCVN 4030-03 and are shown in Table 1. The chemical and compositions of cementitious materials measured by X-ray fluorescence (XRF, S4 Pioneer, Bruker, Karlsruhe, Germany) are given in Table 2. The particle size distribution and the mean particle size of raw materials are presented in Figure 1. The morphology of zeolite particles was observed via scanning electron microscopy (SEM, S-4800, Hitachi, Tokyo, Japan) imaging, as shown in Figure 2. Polycarboxylate-based superplasticizer (SP, ACE388, BASF, Ludwigshafen, Germany) with 35% solid content by weight was used for controlling the workability of HPC mixtures.

### 2.2. HPC Compositions

In order to study the effects of zeolite content and its fineness on the compressive strength and autogenous shrinkage of HPC, six HPC mixtures were proposed which had the same water to binder ratio of 0.18 (without water absorption on sand) and sand/binder (s/b) ratio of 1, while ranging the zeolite content from 0% (control samples) to 15% and the mean particle sizes from 5.6 to 16.7 µm. The water absorption of sand was determined according to Vietnamese standard TCVN 7572-4:06, which revealed a value of 1.15%. This value was used to correct the water required for the hydration. The superplasticizer was used to control the workability of the mixtures, with dosages ranging from 1.20 to 1.65% with respect to the mass of binder. Details of the mix compositions are summarized in Table 3. The binder herein is the total of the cement and zeolites.

### 2.3. Testing Methods

The workability of mixtures was measured using a flow table test. The flowability diameters were between 200 and 230 mm (pursuant to BS 4551-1:1998 [23]). Compressive strength of concrete was determined in accordance with the standard ASTM C109 with the sample size of 50 × 50 × 50 (mm) [24]. The compressive strength was determined using a hydraulic press with a load increment of 2.5 kN/s. The autogenous shrinkage of HPCs was measured on three 25 × 25 × 285 mm samples, with up to 90-day hydration. Mixtures were cast into the moulds and cured at 27 ± 2 °C (close to average ambient temperature in Vietnam and according to the Vietnamese standard TCVN 3105-1993). For each test, a set of 3 samples were prepared. The length of the samples was monitored from the final setting of the mixtures and automatically recorded every 10 min.

The relative humidity test of the HPC sample was determined on a sample of size 100 × 100 × 100 mm. After being cast, a hole with a diameter of 10 mm and a depth of 60 mm was drilled into the sample, in which and RH sensor was placed. The sample surface was covered by a plastic sheet to avoid moisture exchange with the surrounding environment. RH values were recorded regularly during the test.

The shrinkage cracking of HPC was determined by a restrained ring mould in accordance with ASTM C1581-2004, as shown in Figure 3 [25]. The mould was removed after 24 h casting and cured in a climate-controlled chamber (23 ± 2 °C, 50% ± 4% RH). The deformation of the steel ring was then started to be measured. Note that the upper surface of the concrete sample was coated with silicon to ensure the evaporation of water only taking place along its surrounding surface. The experimental setup is shown in Figure 4.

## 3. Results

### 3.1. Workability of HPC Mixtures

As mentioned, the workability of HPC mixtures was controlled by the superplasticizer to obtain the flowability diameter in the range of 200–230 mm. Dependency of the flowability diameter on the dosage of SP is shown in Figure 5. It can be seen that for zeolite with 10.3 µm size, the minimal required SP dosage to ensure the designed workability was obtained at 5% zeolite replacement. After reaching the minimal content, the higher the zeolite replacement content, the larger the SP dosage required. The required SP dosage for the control sample (e.g., without zeolite) was a bit higher than the one for the sample with 5% zeolite replacement, but much lower than the ones for higher zeolite replacements. In addition, with the same 10% zeolite replacement, when the size of zeolite particles increased from 5.6 to 16.7 µm, the required dosage of superplasticizer significantly increased, i.e., 25%. Typically, by adding a small amount of zeolite to the HPC mixture (e.g., 5% in this study), the workability is expected to be improved due to the filling and dispersion effects of very fine zeolite particles, resulting in a higher amount of free water in the paste matrix and thus improving the fluidity of the concrete mixture. However, if the zeolite content continues to increase, due to the large porous structure of zeolite particles as shown in Figure 2, a large amount of water is absorbed into its fine structure, hence reducing the workability of the concrete mixture. This is evident when the zeolite content increases up to 15%, which requires a higher amount of superplasticizer to obtain the designed workability (Figure 5). Furthermore, with a reduced zeolite size, the workability of the concrete mixture could increase because smaller size zeolite requires an intensive grinding, inducing a partial collapse of the porous structure of zeolite particles, thus reducing the amount of absorption water.

### 3.2. Internal Relative Humidity of HPC

Experimental results of the effect of the zeolite content on the relative humidity reduction in HPC samples are shown in Figure 6. It is clear that for the control sample (0% Ze), the internal relative humidity of HPC quickly dropped to a value of 90%, 85% and 80% after 7, 14 and 28 days of curing, respectively. For samples using 5% zeolite, the RH reduction was initially slightly slower than that of the control samples. The RH reduction was then similar to the one of reference samples after 14 days of curing, as seen in Figure 6. When increasing the zeolite content to 10% and 15%, the RH in the HPC sample was slightly reduced in the first 7 days; especially, the RH of concrete with 15% zeolite content was almost stable within this period. While, at 28 days, the RH value was still over 95% but decreased to 86% at the age of 56 days for HPC containing 15% zeolite. Such high rates of RH reduction in samples using 10–15% zeolite compared to the control sample are attributed to the porous structure of zeolite particles. These porous particles can absorb and retain water and are well distributed in the hardened cement paste. When RH in HPC reduces due to the continuous cement hydration process, water retained in these porous zeolite particles will be released and offset some of the water loss due to hydration of cement, thus compensating the reduction of RH in concrete.

### 3.3. Compressive Strength Development

Figure 7 shows the compressive strength of HPC specimens at different ages measured on 3 replicates for each mixture. The compressive strength gain ($f_{c}^{\prime }\left(t\right)$) normalized to 28-day compressive strength of the corresponding specimen (calculated based on the mean values) is shown in Figure 8. For comparison, compressive strength gains predicted from the *fib* 2010 equation are also presented in Figure 8. The compressive strength of HPC tended to increase with the increase in zeolite content of up to 10 %, beyond which it decreased slightly (Figure 8a). This trend was more notable at early ages. For example, when compared to the compressive strength of the reference specimen without zeolite, the specimen with 10% zeolite content developed higher strengths of 149%, 124% and 123% at 7, 28 and 91 days, respectively. Whereas, for the specimen with a higher zeolite content of 15%, the strength developments were a bit lower, i.e., 131%, 110% and 118% at 7, 28 and 91 days, respectively. Note that HPC with 15% zeolite content still exhibited higher compressive strength than the reference specimen without zeolite. It was also observed that with the same zeolite content, the 28-day compressive strength of HPC only decreased slightly with the increase of the particle size of zeolite (Figure 8b). Compared to the reference specimen, the 28-day compressive strength increased by 28.5%, 24.2% and 15.2% for HPC containing zeolites with particle size of 5.6 to 10.3 µm and 16.7 µm, respectively. The strength development of HPC is mainly controlled by the w/b ratio. Despite difference in particle size, the amount of absorbed water in its structure is more or less similar. The difference is only come from the absorbed water on its surface (i.e., the smaller particle size zeolite absorbs more water on its surface). However, the surface-absorbed water is much smaller compared to the structure-absorbed water. Therefore, the particle size of zeolite only slightly affects the compressive strength development. Furthermore, the zeolite particle size could affect the particle packing of the HPC, and for this effect, we only observed a slight increase in compressive strength for samples with a smaller zeolite particle size. This implies that, in the range of investigation, the particle size difference is not significant enough to make a huge impact on the particle packing. In addition, the pozzolanic effect of different zeolite particle sizes on compressive strength is not significant (as shown later in Figure 9b). However, the decreasing rate in the compressive strength with regard to the zeolite particle size became marginal in the later ages. This implies that the smaller particle size of zeolite enhances the compressive strength of concrete at early ages while the larger particle size is favourable in enhancing the strength in the longer ages. The absorbed water in the porous structure of zeolites during mixing can later compensate the RH reduction in concrete at early ages. Despite this compensation effect, the RH still decreased with ages, as seen in Figure 6.

The compressive strength development of HPC was represented by a parabolic shape (Figure 8), thereby indicating that the increasing rate of compressive strength gradually decreased with ages. However, it is interesting to point out that the addition of the high zeolite content, i.e., 15%, significantly improved the compressive strength at later ages, from 28 days to 90 days. The strength gain ratios of HPC with different zeolite contents normalized to the 28-day strength generally ranged between 0.59 and 0.73 at 3 days and 0.80 and 0.88 at 7 days. The corresponding ratios were from 1.08 to 1.18 at 91 days. The strength gain ratios at early ages tended to decrease with the increase in zeolite content (Figure 8a). However, all HPC specimens with zeolite content exhibited higher strength gain ratios at early ages than the one of the reference specimen and showed a higher strength development than the predicted results using the *fib* 2010 model. Thus, the compressive strength development of HPC with zeolite content can be conservatively assessed by using the empirical equations specified in the *fib* 2010 model. In contrast, the zeolite particle size insignificantly affected the compressive strength development of HPC (Figure 8b). Parabolic curves of compressive strength development obtained for specimens with different zeolite particle sizes were close to each other. These specimens also displayed a slightly higher strength gain ratio than the predictions by the *fib* 2010 model.

### 3.4. Pozzolanic Effect of Zeolite in the HPC system

The high-performance concrete is typically made from binders composed of cement as the main binder and mineral admixtures as supplementary cementitious materials. Therefore, the contribution of the binders to the strength of concrete is originated from both the cement hydration and the secondary pozzolanic reaction of the mineral admixtures with portlandite produced during the cement hydration. Besides, the water-reducing effect and filler effect of mineral admixtures also contribute to the strength development of concrete. All of these contributions to the concrete strength are defined as the pozzolanic effect.

In fact, each mineral admixture will differently contribute to the absolute concrete strength development, which makes it impossible to directly compare the pozzolanic effect of different mineral admixtures. Thus, Xincheng [26] proposed to quantify the pozzolanic effect by normalizing the concrete strength to the cement content used in concrete, called the specific strength of cement content in concrete, or in short, the specific strength of concrete. The specific strengths of concrete with ($R_{sa}$) and without a mineral admixture ($R_{sc}$) are simply defined as follows:(1)$$R_{sa}=\;R_{a}\;/\;q_{o};\;R_{sc}=\;R_{c}\;/\;100$$
where $R_{a}$ and $R_{c}$ (MPa) are the absolute concrete strength with and without mineral admixture respectively, and $q_{o}$ (%) is the mass percentage of cement in the concrete containing mineral admixture. The difference between these two specific strengths is considered as the strength development due to pozzolanic effect ($R_{sp}$, MPa)
(2)$$R_{sp}=\;R_{sa}-\;R_{sc}$$

The strength contribution percentage of pozzolanic effect ($P_{a}$, %) to the total strength development is then defined as:(3)$$P_{a}=\;\frac{R_{sp}}{R_{sa}}\;\times 100\;\left(\%\right)$$

In this study, the zeolites with high SiO_2_ content (>60%, Table 1) and fine particle size are expected to have a larger pozzolanic effect. The contribution of pozzolanic effect to the total strength of HPC is then calculated using Equation (3) and shown in Figure 9.

Note that in order to estimate the standard error, we used the mean values of *R_c_.* It is clearly seen that the contribution of zeolite to the compressive strength of HPC due to the pozzolanic effect was significant at early ages, i.e., from 3 to 7 days, and became less significant at later ages, i.e., after 28 days, especially for the sample with only 5% zeolite replacement. It implies that the pozzolanic effect of zeolites was still active for both early and later ages thanks to its fine but wide range of particle size distribution. This also means that the portlandite content in HPC is sufficient, even for later ages, to contribute to the pozzolanic reactions. Note that for other porous materials, using an internal curing agent such as lightweight aggregates [27,28] is typically effective with respect to the pozzolanic effect at early ages because of its coarse pore structure. The particle size influence is also observed in this study, as illustrated in Figure 9b. The smaller the particle size, the higher the pozzolanic effect. With a smaller size, the zeolite is expected to increase the pozzolanic reaction because of its higher specific surface area. Furthermore, water as a reactive medium can be equally distributed into the tiny pore network to reduce the local RH drop in concrete, which potentially reduces the pozzolanic reaction. This is especially important for HPC with a very dense microstructure that prevents the redistribution of water to surrounding regions due to its low permeability [29]. On the other hand, if the particle size of zeolite is significantly reduced by grinding, the porous structure of zeolite particles is supposed to be partly collapsed, similar to the case of rice husk ash [30,31]. This would reduce the effectiveness of the pozzolanic effect because of less absorbed water in the zeolite structure. Therefore, the pozzolanic effect is clearly seen at early ages, but is much reduced at later ages. This phenomenon was also observed in this study; however, we did not see a significant difference between 3 investigated particle sizes, implying that the extent of collapse was not much different between these 3 particle sizes.

### 3.5. Autogenous Shrinkage

Figure 10 shows the autogenous shrinkage strains ($\varepsilon_{Ash}$) measured in the 3 prepared specimens for each case. The shrinkage mainly occurred within the first 14 days after casting and was followed by a considerably slow shrinkage rate, meaning that the autogenous shrinkage of HPC exhibited an exponential distribution, as commonly observed in normal-strength concrete [32]. The amount of $\varepsilon_{Ash}$ at 14 days corresponded to 92% and 97% of that measured at the ages of 28 and 56 days respectively, indicating that the autogenous shrinkage of HPC mostly occurred at the early ages. The autogenous shrinkage tended to decrease with the increase in zeolite content. In comparison with the $\varepsilon_{Ash}$ of the reference specimen measured at 56 days, HPC specimens with zeolite exhibited much lower shrinkage with a reduction in strain values by 23%, 45% and 56% for specimens with 5%, 10% and 15% zeolite content, respectively. In addition, the slopes of the increase in $\varepsilon_{Ash}$ at early ages decreased with the increase in zeolite content. The reference specimen displayed a sharp increasing rate in $\varepsilon_{Ash}$ within the first 3 days, whereas the rapid increasing slope observed in the specimen with 15% zeolite content appeared until the age of 12 days, before following a slow shrinkage development. The particle size of zeolite also significantly affected the increasing rate of $\varepsilon_{Ash}$ at early ages and the ultimate $\varepsilon_{Ash}$ of HPC. The larger the zeolite content, the better the shrinkage reduction. The $\varepsilon_{Ash}$ at 56 days decreased by approximately 22% with the zeolite particle size increasing from 5.6 to 16.7 µm. The period in which a sharp shrinkage increase occurred was prolonged from 10 days to 21 days with an increase in zeolite particle sizes from 5.6 to 16.7 µm.

The reduction in the shrinkage of concrete with zeolite is associated with an extremely low w/b ratio and the fine pore system of HPC. With the progress of the hydration process, the amount of water in the pore system decreases, thus, reducing the relative humidity in concrete and causing self-desiccation in the capillary structure of hardened paste matrix inducing higher shrinkage deformation in the concrete. However, when using zeolite, the RH reduction due to water involvement in the process of cement hydration could be compensated. In addition, with a lower w/b ratio in the particular case of HPC, due to extremely dense microstructure, the water from reservoirs is difficult to transfer to the surroundings. Therefore, in order to increase the capacity of internal curing, the water reservoirs should be better distributed and divided into smaller filled water pores, e.g., using smaller porous zeolite particles. However, it should be noted that at a certain grinding extent, the zeolite structure can be collapsed, as discussed above. As a result, the amount of absorbed water in zeolite particles will be reduced, thus decreasing the effectiveness of using zeolite in mitigating the autogenous shrinkage of HPC with a smaller zeolite particle size. In this study, though the collapse of zeolite particle structure did not significantly influence the strength development, it seems to be very important for the autogenous shrinkage, which resulted in a larger shrinkage reduction with the smaller zeolite particle size, as proven in Figure 8b.

### 3.6. Crack Resistance

In this study, the possibility of cracking resistance due to the shrinkage of HPC containing different zeolite contents was carried out. Experimental results on deformation of steel rings due to shrinkage of concrete rings are shown in Figure 11. It appeared that the cracks of the reference sample (0% Ze) occurred after 1.3 days of casting. For HPC samples with zeolite, the cracks occurred at considerably later ages of 3.5, 6.8 and 12.5 days for 5%, 10% and 15%, respectively. Therefore, the use of zeolite plays an important role in limiting cracks, thereby slowing the process of cracking on concrete structures. The quick appearance of cracks in the reference sample can indeed be explained by the large shrinkage of HPC. When the amount of water in concrete is lost due to the hydration of cement, known as the self-desiccation process, in the hardened cement matrix, and due to evaporation of water to the environment, the relative humidity in concrete is decreased. Such dehydration process will cause stress in the pores of the concrete structure, which is considered as the total surface tension on the meniscus surface of water in the pore system of concrete, thereby causing shrinkage of concrete. For HPC, the self-desiccation process is stronger at the early ages compared to HPC using zeolite, as is clearly shown in Figure 6, resulting in a shorter period of cracking occurrence.

## 4. Modelling the Autogenous Shrinkage of HPC Containing Zeolites

The autogenous shrinkage strains of HPC is typically characterised by an exponential distribution that has both acceleratory and slow flow periods for the increasing rate of shrinkage with age, as shown in Figure 10. Thus, the autogenous shrinkage strain ($\varepsilon_{Ash}\left(t\right)$) of concrete at a given time (*t*) is commonly expressed by the product of the ultimate autogenous shrinkage strain ($\varepsilon_{Ash\propto }$) and an exponential time function ($T_{A}\left(t\right)$), as expressed in the following form [32,33]:(4)$$\varepsilon_{Ash}\left(t\right)=T_{A}\left(t\right)\cdot \varepsilon_{Ash}$$

### 4.1. Time Function

The increasing rate in autogenous shrinkage of concrete with age depends on the hydration progress of cement. In general, cement hydrates rapidly up to a certain period, beyond which the hydration rate gradually slows down preventing the moisture movement in cement paste matrix Thus, the *fib* 2010 model identifies the time function for autogenous shrinkage as a negative exponential function as follows:(5)$$T_{A}\left(t\right)=1-\exp\left(\alpha_{1}\sqrt{t}\right)$$
where $\alpha_{1}$ is the coefficient to represent the slope of the time function, which is set to be –0.2 in the *fib* 2010 model on the basis of the test data of normal strength concrete. However, the rate of increment of autogenous shrinkage is affected by various conditions, including curing temperature, cement content, fineness and type of cementitious materials. This implies that the HPC with zeolite would exhibit a different shape of time function with the mitigation of autogenous shrinkage, as previously presented in Figure 10.

To determine the value of $\alpha_{1}$ in HPC with zeolite, a regression analysis was performed using the present test dataset. The value of $\varepsilon_{Ash}$ can be obtained from $\varepsilon_{Ash}\left(t_{1}\right)$ of concrete at a specified point ($t_{1}$) and $T_{A}\left(t\right)$. When *t* approaches infinity, the exponential time function will converge on unity. Because of the fact that it is impractical to perform an extremely long-term shrinkage measurement, $\varepsilon_{Ash}$ can be calculated from $\varepsilon_{Ash}\left(t_{1}\right)$ with the adoption of a best-fitting regression procedure which uses a predictive model for shrinkage time history. Thus, $\varepsilon_{Ash}$ is determined from $\varepsilon_{Ash}\left(t_{1}\right)/T_{A}\left(t_{1}\right)$ when the experimental record is given at $t_{1}$. Overall, $\varepsilon_{Ash}\left(t\right)$ can be calculated from the following equation [33]:(6)$$\varepsilon_{Ash}\left(t\right)=\;\frac{T_{A}\left(t\right)}{T_{A}\left(t_{1}\right)}\varepsilon_{Ash}\left(t_{1}\right)$$

The present study selected 14 days as the specified time, $t_{1},$ because most specimens showed the inflection time to convert from the initial acceleratory period to flow period in the autogenous shrinkage strain curve at approximately 14 days. By using the present test dataset normalized by Equation (4), the value of $\alpha_{1}$ in Equation (5) was calculated in each specimen. The value of $\alpha_{1}$ was affected by the content and fineness of zeolite. From the regression analysis of $\alpha_{1}$ determined in each specimen, the following equation for shrinkage time history could be obtained (Figure 12):(7)$$\alpha_{1}=1.55{\left[1/\left(1-R_{ze}\right)\right]}^{-4}{\left(P_{s}/P_{o}\right)}^{-0.4}$$
where $R_{ze}$ is the ratio of zeolite content relative to cement content by weight, $P_{s}$ (µm) is the mean particle size of zeolite and $P_{o}$ (= 1 µm) is the reference value for particle size. Note that $\alpha_{1}$ has a positive sign compared to negative sign in the *fib* 2010 model.

### 4.2. Ultimate Autogenous Shrinkage Strain ($\varepsilon_{Ash\infty }$
)

According to the definition of time function, $T_{Ash}\left(t_{\infty }\right)$ is equal to 1. Hence, $\varepsilon_{Ash\infty }$ in Equation (8) for the specified time of 14 days can be written in the following form:(8)$$\varepsilon_{Ash\infty }=\frac{1}{T_{Ash}(t_{14})}\varepsilon_{Ash}(t_{14})\;=\;\frac{\varepsilon_{Ash}\left(14\right)}{\left[1-exp(-\alpha_{1}\sqrt{14})\right]}$$

From Equations (7) and (8), $\varepsilon_{Ash\infty }$ can be determined providing that the autogenous shrinkage strain at the specified time of 14 days is given. The value of $\varepsilon_{Ash}\left(14\right)$ was formulated from regression analysis using the present test data considering the influencing parameters aforementioned in Figure 10. As a result, $\varepsilon_{Ash}\left(14\right)$ for HPC with zeolite can be expressed as follows and is graphically shown in Figure 13:(9)$$\varepsilon_{Ash}(14)=1992{\left[1/\left(1-R_{ze}\right)\right]}^{-5.8}{\left(P_{s}/P_{o}\right)}^{-0.58}$$

Finally, the autogenous shrinkage behaviour of HPC with zeolite can be estimated using the following equation:(10)$$\varepsilon_{Ash}\left(t\right)=\frac{1-\exp\left(-\alpha_{1}\sqrt{t}\right)}{1-\exp\left(-\alpha_{1}\sqrt{14}\right)}\cdot 1992{\left[1/\left(1-R_{ze}\right)\right]}^{-5.8}{\left(P_{s}/P_{o}\right)}^{-0.58}$$

### 4.3. Comparison of the Proposed Model vs. the Fib 2010 Model 

Figure 14 displays the comparisons between experimental results and the prediction of autogenous shrinkage strains of HPC containing zeolites using the proposed model and the *fib* 2010 model. Note that due to limited data available for the autogenous shrinkage of HPC, especially HPC containing zeolite, only the current dataset was used to examine the validation of the proposed model and the *fib* 2010 model. The prediction obtained from the *fib* 2010 model tends to underestimate the autogenous shrinkage behaviour of HPC. For the reference specimen and specimens containing 5% zeolite with the particle size of 10.3 µm and with 10% zeolite with the particle size of 5.6 µm, the *fib* 2010 model exhibits significantly lower shrinkage compared to the measurements. This underestimation was alleviated with the increase in zeolite content and particle size of zeolites. The underestimation of shrinkage of the *fib* 2010 model is somehow unexpected because the zeolite presented in HPC reduces the shrinkage. This implies that higher shrinkage due to low w/b ratio and high paste volume of HPC is still the dominated process compared to zeolite reduction shrinkage, and thus largely contributes to the ultimate shrinkage strain. The closest predictions are obtained for HPC mixtures with 10% zeolite with the mean particle size of 16.7 µm and 15% zeolite with the mean particle size of 10.3 µm. The mean and standard deviation of normalised root-mean-square errors (NRMSE) calculated from the comparison between the experiments and predictions by *fib* 2010 for those cases are 0.39 and 0.22, respectively. While, the proposed model displays a better agreement with the test results, with mean and standard deviation values of NRMSE being 0.18 and 0.16, respectively. This high accuracy is attributed to the fact that the regression analysis was done using the current data only. When applying the proposed model directly to other HPC, the accuracy may be lower due to differences in raw material properties and mixture compositions. In that case, the model needs to be recalibrated for a better prediction. For all the specimens, except for HPC mixture containing 10% zeolite with 5.6 µm particle size, consistent predictions are obtained with regard to acceleratory and slow flow periods for the increasing rate of autogenous shrinkage with age. Note that the proposed model is only calibrated at the specific time of 14 days, implying that the proposed model is able to predict the shrinkage behaviours (reflecting via the time function and ultimate shrinkage strain) of HPC with and without zeolites at various ages.

## 5. Conclusions

In this work, the effects of zeolites on the autogenous shrinkage of high-performance cement-based concrete were evaluated on various HPC mixtures containing up to 15% zeolite replacement of cement with three mean zeolite particle sizes of 5.6, 10.3 and 16.7 µm. Based on a rich experimental dataset and with the support from the modelling approach, a number of key conclusions can be drawn as follows:
The addition of zeolites up to 10% resulted in beneficial effects in both the flowability and compressive strength of high-performance concrete. Higher zeolite content could reduce the workability and the compressive strength, though it was still higher than the one of HPC without zeolite.Zeolites with a porous structure helped to reduce the drying of HPC due to self-desiccation and continuous hydration of a low w/b ratio cementitious matrix. The higher the zeolite content, the smaller the RH reduction in HPC. Consequently, the autogenous shrinkage tended to decrease with the increase in zeolite content and its particle size because of higher water absorption capacity in its structure. In addition, the rate of the autogenous shrinkage development at early ages decreased with the increase in zeolite content. A sharp increasing rate in autogenous shrinkage was observed within a short period of 3 days, while it was prolonged up to 12 days for the sample with 15% zeolite.The shrinkage reduction resulted in a lower potential of shrinkage cracking of HPC containing zeolites. The first shrinkage cracks occurred at 1.3 days for the reference sample, whereas the shrinkage cracks only appeared at 6.8 days and 12.5 days for HPC mixtures with 10% and 15% zeolite, respectively.An improved model for HPC containing zeolites has been proposed by introducing new parameters accounting for the zeolite content and its particle size. It appeared that the proposed model was able to capture the autogenous shrinkage behaviour of HPC with or without zeolite, while the *fib* 2010 model underestimated the autogenous shrinkage of HPC containing less than 10% zeolite replacement. Note that when applying the proposed model to other HPC with different raw material properties and compositions, the model may need to be recalibrated to increase the accuracy.

## Figures and Tables

**Figure 1 materials-13-03773-f001:**
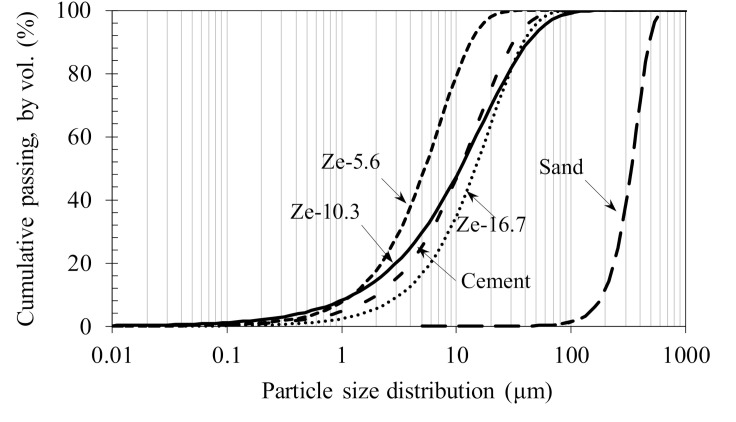
Particle size distribution of raw materials used in this study.

**Figure 2 materials-13-03773-f002:**
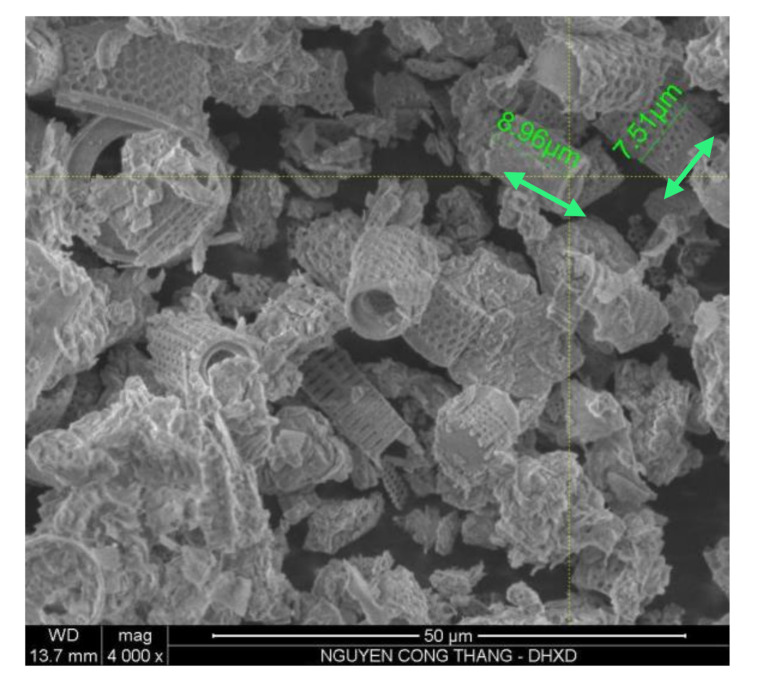
Scanning electron microscopy (SEM) image of zeolite particles.

**Figure 3 materials-13-03773-f003:**
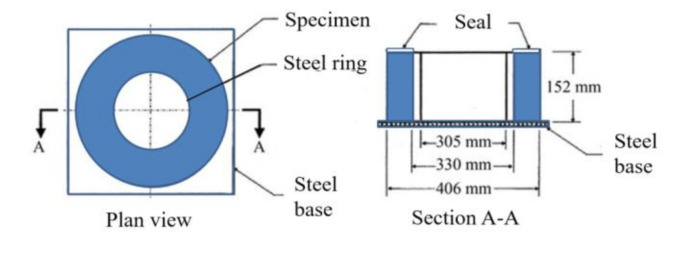
HPC restrained shrinkage ring test model.

**Figure 4 materials-13-03773-f004:**
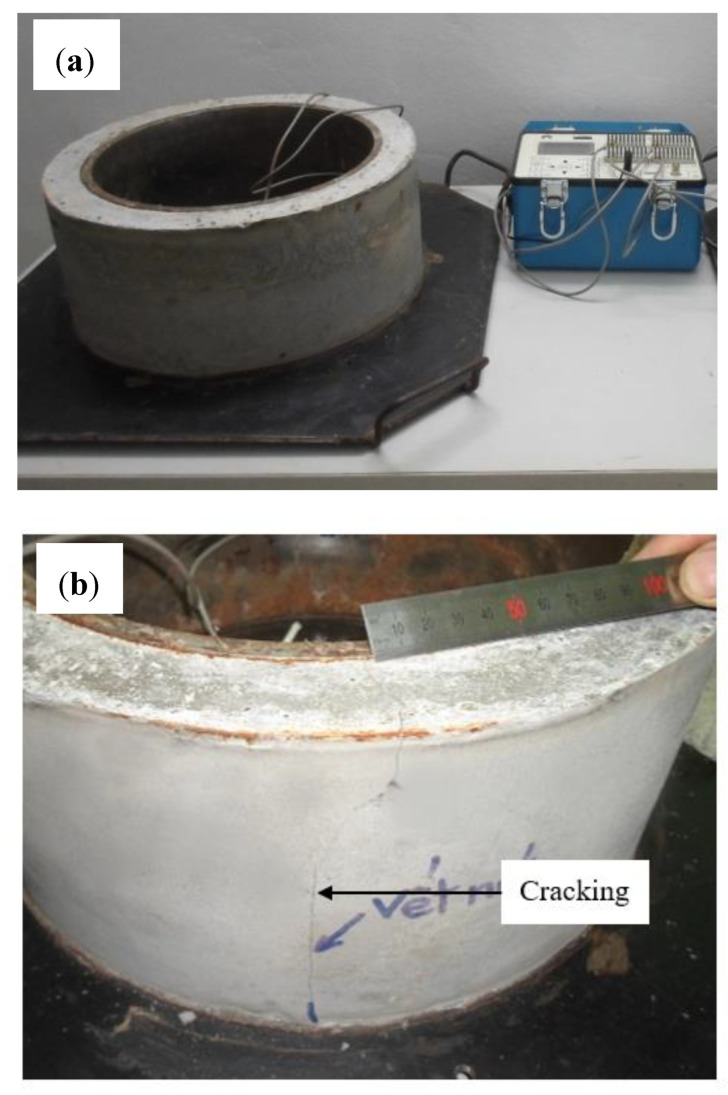
Restrained shrinkage test for HPC samples: (**a**) experimental setup and (**b**) crack formation.

**Figure 5 materials-13-03773-f005:**
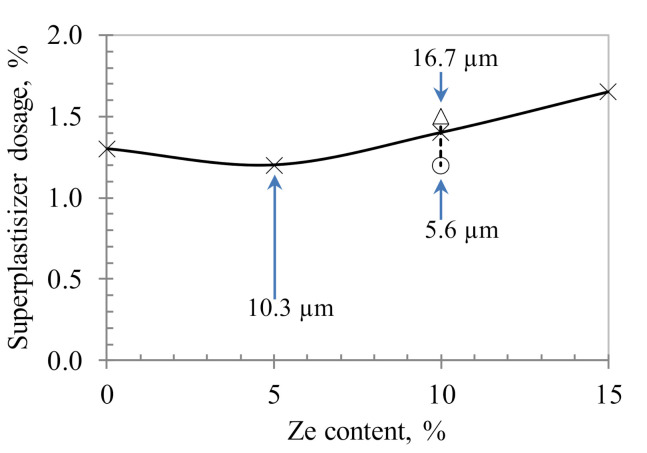
Relationship between the dosage of superplasticizer and the zeolite replacement for HPC mixture with water/binder ratio of 0.18 and flowability diameter ranging from 200–230 mm.

**Figure 6 materials-13-03773-f006:**
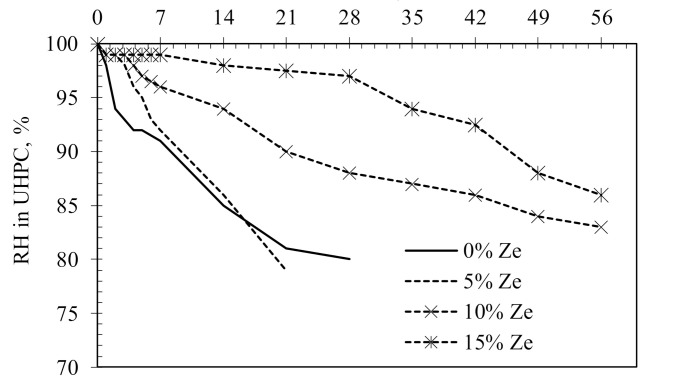
The internal relative humidity (RH) evolution over curing time in HPC.

**Figure 7 materials-13-03773-f007:**
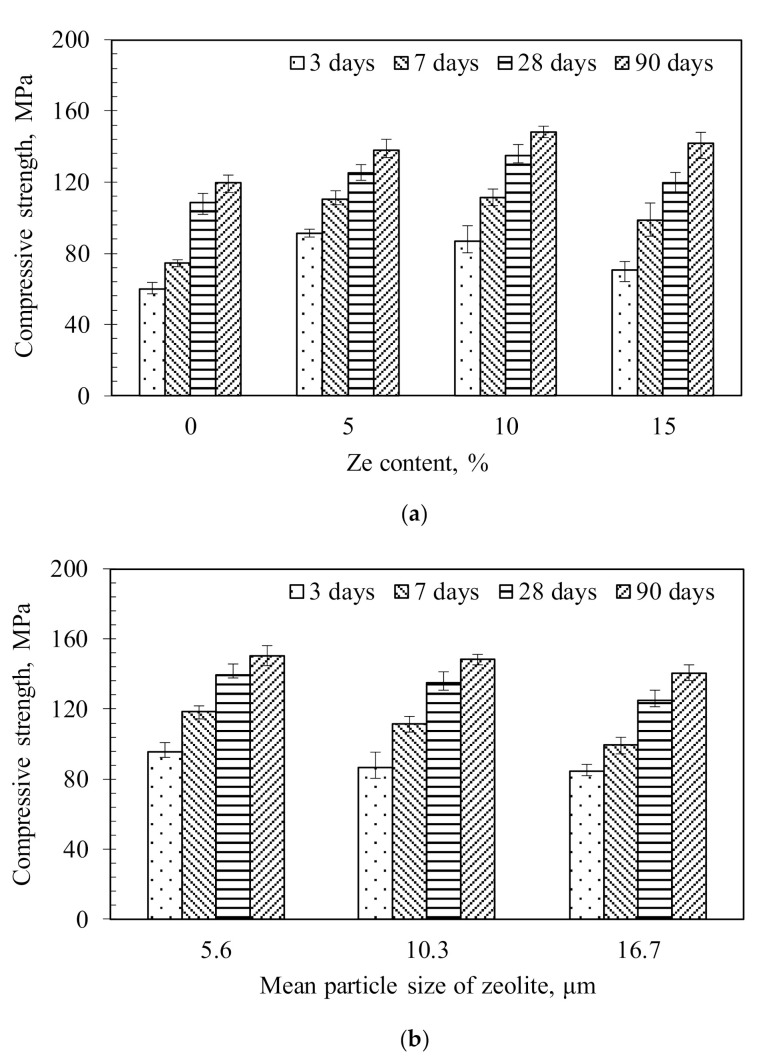
Effect of the zeolite content (**a**) and the particle size of zeolite (**b**) on the compressive strength of HPC specimens—tests were performed on 3 replicates.

**Figure 8 materials-13-03773-f008:**
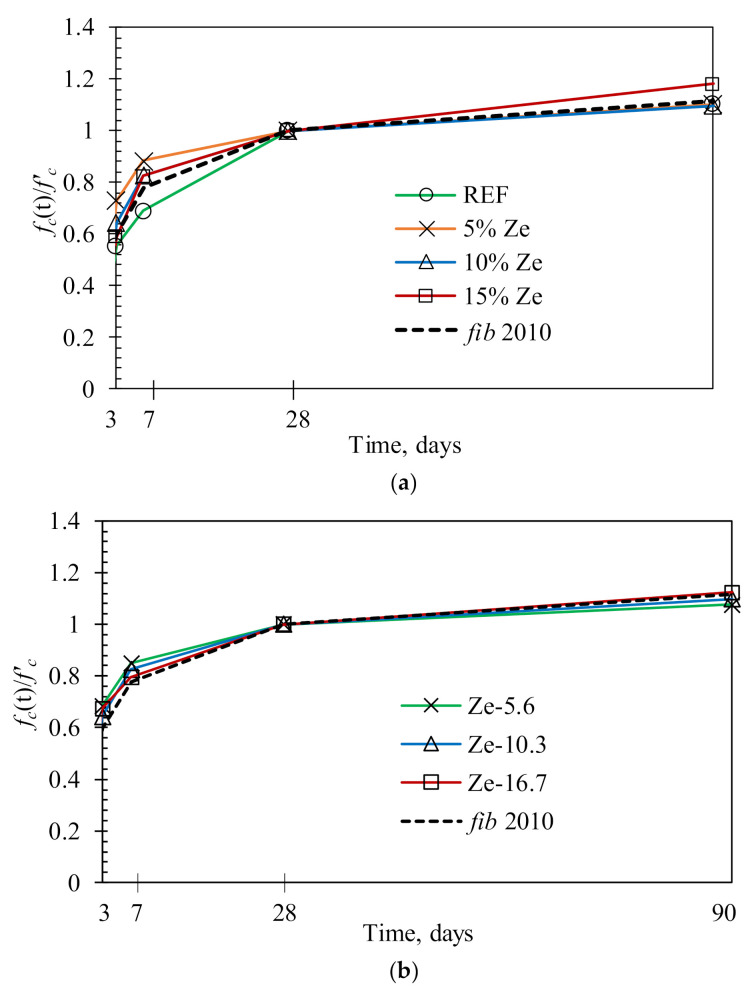
Effect of the zeolite content (**a**) and the particle size of zeolite (**b**) on the compressive strength development of HPC specimens.

**Figure 9 materials-13-03773-f009:**
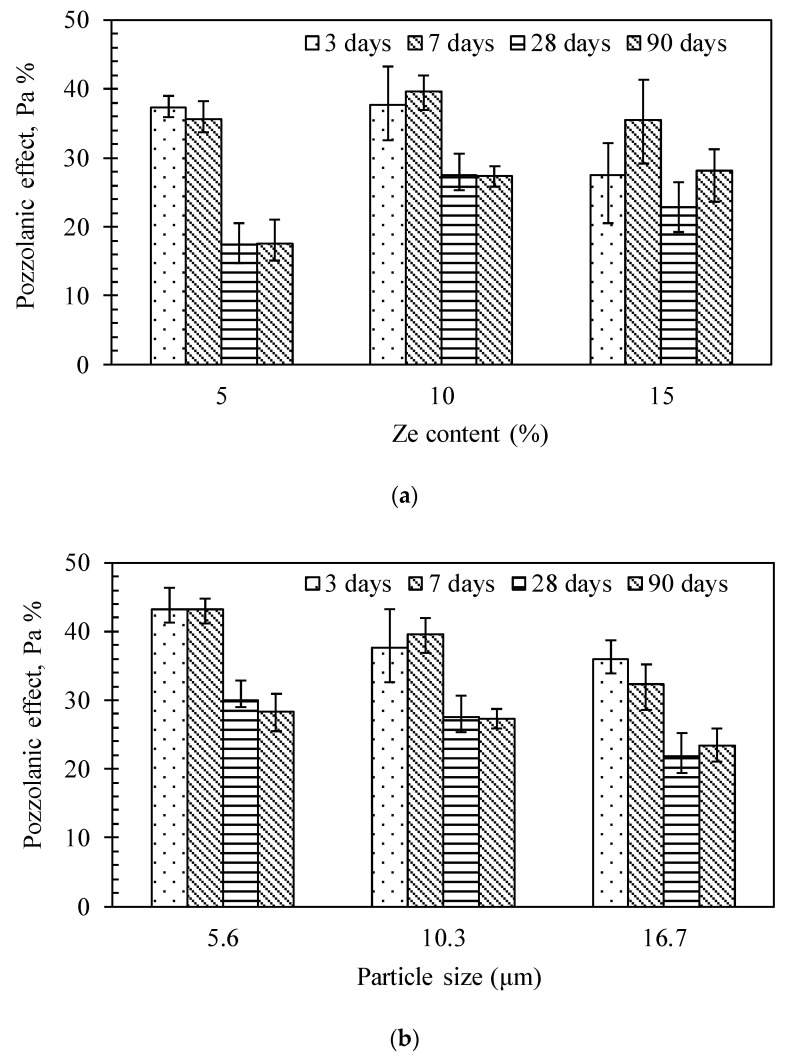
Effect of the zeolite content (**a**) and the particle size of zeolite (**b**) on the pozzolanic effect of zeolite in the HPC system.

**Figure 10 materials-13-03773-f010:**
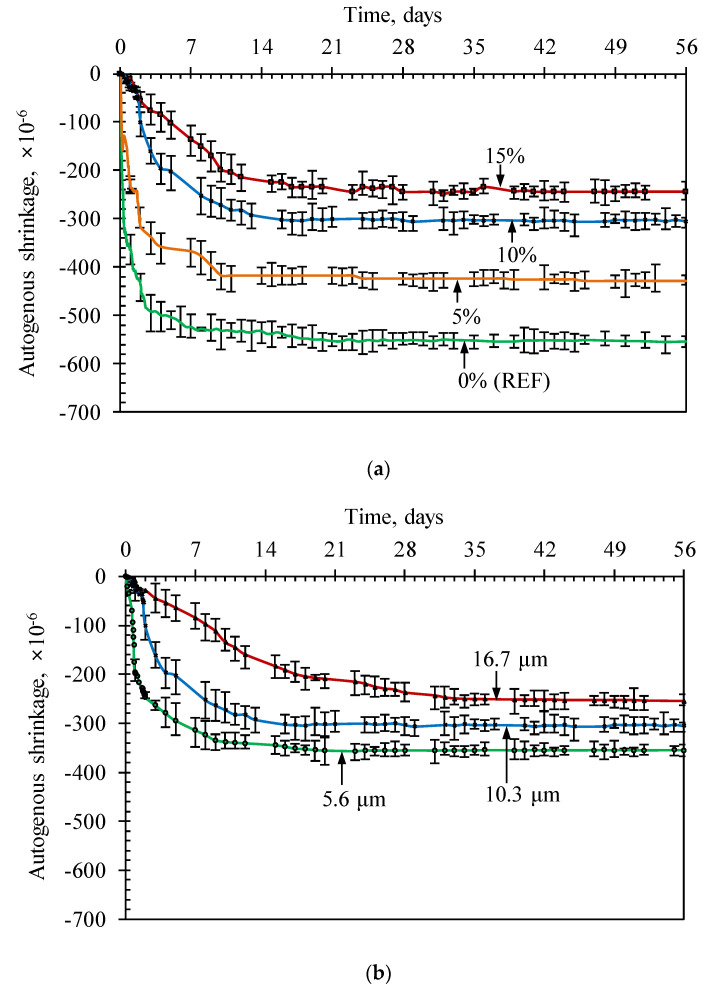
Effect of the zeolite content (**a**) and the particle size of zeolite (**b**) on autogenous shrinkage of HPC specimens—measurements were performed on 3 replicates.

**Figure 11 materials-13-03773-f011:**
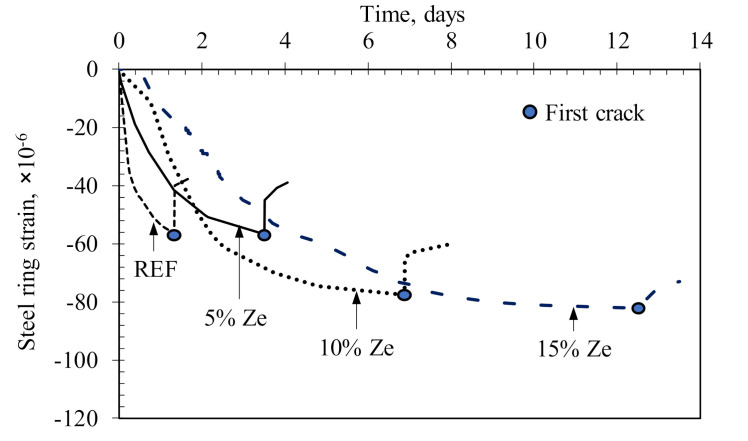
Deformation of steel rings in restrained shrinkage experiment.

**Figure 12 materials-13-03773-f012:**
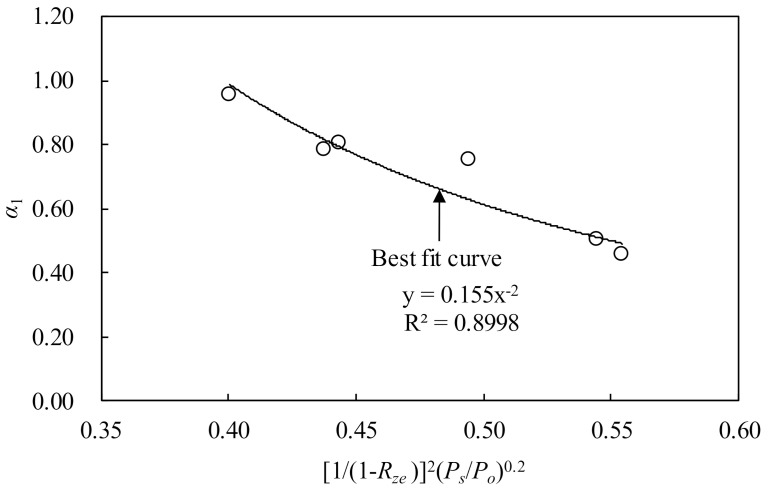
Regression analysis of *α*_1_ in time function.

**Figure 13 materials-13-03773-f013:**
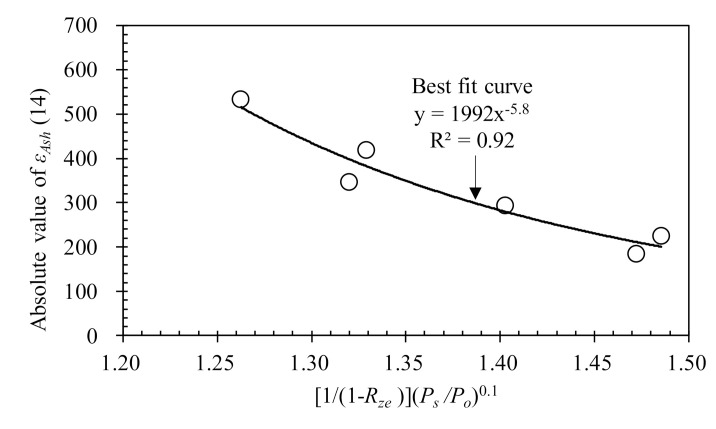
Regression analysis of $\varepsilon_{Ash}\left(14\right).$

**Figure 14 materials-13-03773-f014:**
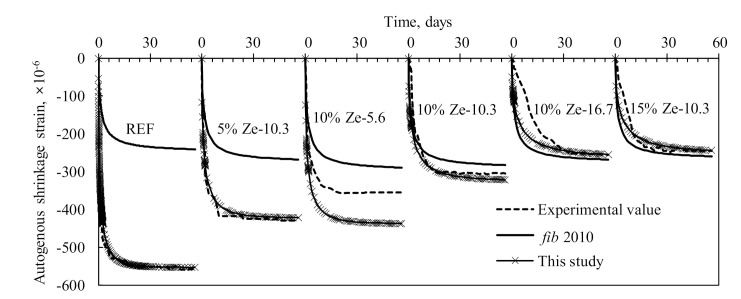
Comparisons between experimental and predicted autogenous shrinkage strains using the *fib* 2010 model and proposed model.

**Table 1 materials-13-03773-t001:** Physical properties of cementitious materials.

Properties	Cement	Ze-5.6	Ze-10.3	Ze-16.7
Density, g/cm^3^	3.15	2.18		
Mean particle size, μm	13.9	5.6	10.3	16.7

**Table 2 materials-13-03773-t002:** Chemical compositions of cementitious materials.

Materials	SiO_2_	Fe_2_O_3_	Al_2_O_3_	CaO	MgO	Na_2_O	K_2_O	SO_3_	L.O.I
Cement, wt.%	35.58	0.68	12.76	42.82	7.67	0.24	0.25	N/A	N/A
Zeolite, wt.%	60.25	1.9	18.45	4.11	1.12	1.66	1.18	N/A	11.33

**Table 3 materials-13-03773-t003:** Mix composition of high-performance cement-based concrete (HPC) specimens.

No.	w/b	s/b	Ze, wt.% by Binder	Solid SP, wt.% by Binder	S, kg/m^3^	Ze, kg/m^3^	C, kg/m^3^	SP, kg/m^3^	Water, kg/m^3^	Mean Particle Size of Ze, μm
1	0.18	1	0	1.30	1143	0	1143	42.5	191	10.3
2	0.18	1	5	1.20	1139	56.9	1082	39.0	193	10.3
3	0.18	1	10	1.40	1134	113.4	1021	45.4	188	10.3
4	0.18	1	15	1.65	1130	169.5	961	53.3	182	10.3
5	0.18	1	10	1.20	1134	113.4	1021	38.9	192	5.6
6	0.18	1	10	1.50	1134	113.4	1021	48.6	186	16.7

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
