# Peer review of "Effect of Zeolite on Shrinkage and Crack Resistance of High-Performance Cement-Based Concrete"

_materials, 2020, doi:10.3390/ma13173773_

Round 1

Reviewer 1 Report

In the article: “Effect of Zeolite on Shrinkage and Crack Resistance of High Performance Cement-Based Concrete” authors examined the effects of zeolite addition on the shrinkage of high performance cement mortars.

The article is written reasonably well, the description of the performed research and analysis is good, the conclusions are supported by obtained results.

Below are my remarks, correction of which, as I believe, will improve this article even more:

- Technically the subject of the performed research by Authors is a mortar, not concrete, therefore I would suggest replacing the word concrete with the word mortar or composite when appropriate (including title).

Concrete is composite of fine and coarse aggregate and cement paste, while mortar contains only fine aggregates and cement paste.

- Line 56, concrete members -> mortar components ?

- Line 94: “The chemical and physical compositions of cementitious materials measured by X-ray fluorescence (XRF) are given in Table 1 and Table 2, respectively.”

Physical properties? Were they measured by XRF as well?

- Tab. 1: The meaning of horizontal lines (-) is not clear. No data or 0?

- Fig. 1: Something wrong with arrows.

- Fig. 2: The arrows near the green numbers/dimensions are barely visible, should be thickened.

- Line 111, “% by binder” might be described in the text as well as it is used in Tab. 3.

- Tab. 3: Check calculations of w/b. Based on the provided data, If I am not mistaken, water without considering water in SP, w/b is 0.16. If that water from SP is taken into account then w/b is 0.19.

- Inconsistency in naming of SP. “SP, wt.% by binder” in Tab. 3 should probably be named “solid content in SP, wt.% by binder”

- Fig. 8. Showing point (0,0) does not make any sense (was it measured ?), unless Authors will prove otherwise.

- Fig. 10 and Fig. 14. Negative shrinkage is expansion. Values at the vertical axis should be positive.

- 189-192. The sentence is not clear. Please rewrite it to be more understandable.

- 196. The description here is not clear. Should be extended to contain a precise description of results presented in Fig. 8b. Values given here are correct only for 28 days?

- 337. “For HPC, the self-desiccation process is stronger at the early ages compared to HPC using zeolite as clearly shown in Error! Reference source not found., resulting in a shorter period of cracking occurrence.”

Probably something went wrong with reference linking.

- 344. Parabolic or exponential as in line 350?

- eq. 5, alpha here is taken as -0.2, but in later considerations e.g. eq. 8 or eq. 10 alpha is assumed positive with a minus sign before. It should be clarified.

- 373. Ps – mean particle size. Was that given in meters?

417 - ...up to 15% zeolite replacement…

replacement of cement? It is written in line 90 already, but in my opinion, it should be repeated here as well for those reading only conclusions.

Author Response

Thanks for your useful comments and suggestions. Please see the attachment.

Reviewer 2 Report

The study presents a timely and well presented investigation of zeolite addition to HPC. It is well motivated and the set of experiments was well thought out. There are a few key points of improvement:

  1. There are a number of grammar issues throughout the manuscript that should be corrected. 
  2. Various points imply an improvement to durability, which is a broad concept which also incorporates permeability. Could any implications on permeability be discussed, based on the existing results? 
  3. The manuscript is wordy and could be more concise. This could be improved by thinking of ways to avoiding repeated statements. For example, the fact that the zeolites absorb water and that it impacts RH is mentioned many times. The effect of degree of grinding is also mentioned too many times. 
  4. Is there an explanation as to why zeolite particle size insignificantly affects the compressive strength development?
  5. There is a reference error on page 13.
  6. The regression equation lacks theoretical basis, which is okay for now and may be a point of future research. However, since it is practically a fitting of the fib model to your data, it is not too valuable to provide a 'comparison' between the two. The comparison section mainly serves the purpose of showing how well you were able to fit your own data so it would be better to motivate section 4.3 as an argument of how well your model can capture your own data. 
  7. The greatest concern about this study is that there is no indication of how repeatable the results are. Cement and concrete are random materials which means that every test will be slightly different. Thus it is correct scientific practice to conduct repeated tests for each case and show the average and scatter of each. A single shrinkage curve or ring test is not convincing. While the data looks very clean, it is hard to accept new results for tests that were only conducted a single time. An average of a number of samples with an indication of the scatter would be extremely valuable to assess all the results and how significant the stated relationships in the conclusions are.

Author Response

(The authors gave the same response as above.)
